# How Did We Get Here: The Best Vaccines Ever Facing the Highest Public Hesitancy?

**DOI:** 10.3390/vaccines11081323

**Published:** 2023-08-04

**Authors:** Catterina Ferreccio

**Affiliations:** Advanced Center for Chronic Diseases (ACCDIS), School of Medicine, Pontificia Universidad Católica de Chile, Av. Diagonal Paraguay 362, Santiago 8330077, Chile; cferrecr@uc.cl

**Keywords:** vaccine hesitancy, intellectual property, HPV vaccines, COVID-19 vaccines

## Abstract

mRNA vaccine technology is the most interesting final product of decades of research. This new platform for public health is simple to transfer to low-income countries and can be used against diverse agents, including cancer. It is environmentally clean, relatively low-cost, and does not use animals for its production. Most importantly, mRNA vaccines have been highly efficacious in avoiding serious disease and death from COVID-19. Yet, at the highest point of the pandemic, many voices, including some from prominent positions, opposed their use. Similarly, the Human Papillomavirus (HPV) vaccines, which are highly effective, very safe, and probably confer long life protection against its HPV types, faced strong parents’ hesitancy. Vaccine hesitancy has been the subject of extensive research, focusing primarily on factors associated with the public, the political environment, and messaging strategies. However, the issue of unfair worldwide access to the COVID-19 vaccines has recently sparked significant debate about the vaccine industry’s role. Recent data demonstrated that the system’s perceived unfairness with the masses is behind the growing populist anti-vaccine movements worldwide. The association between populism and antivaccine attitudes has been reported at country and individual levels. The anti-science attitudes behind vaccine hesitancy emerge when the scientist is not found credible due to the suspicion that they had monetary investments in pharmaceutical companies. Here, I argue that the obscurity of the vaccine market, but also its unfairness, are important factors contributing to vaccine hesitancy. The purpose of this commentary is to stimulate a review of current market regulations and to improve its transparency and fairness, particularly in the context of public health emergencies. By doing so, a new pandemic would find us better prepared. The general population and much of the healthcare community often ignore the years of dedicated work and substantial public funding that enabled the discovery and design of vaccines. Conversely, pharmaceutical companies often over-emphasize their investments in research and development. A decade ago, Marcia Angell provided a detailed breakdown of pharmaceutical expenses, revealing that marketing and administration costs were 2.5 times higher than research and development expenses; recently, Olivier Wouters confirmed the high expenditures of the pharmaceutical industry in lobbying and political campaign contributions. In this commentary, I will present the cases of HPV and COVID-19 vaccines as examples of when vaccines, instead of being public health goods, became market goods, creating large inequities and health costs. This failure is a structural cause behind more ideological vaccine hesitancy, less studied so far.

## 1. Introduction

Most reports related to vaccine hesitancy have approached it from individual subjective factors, consequently recommending individual-based interventions to increase vaccine acceptance, such as the use of commercial and social marketing principles [1], identifying vaccine hesitancy population subgroups [2], or improving health communication using novel tools and strategies [3]. Only one study included in its Hesitancy Matrix “perception of the pharmaceutical industry” as one of the contextual influences in vaccine hesitancy [4]. A review of vaccine hesitancy interventions from 1996 to 2013 found that targeting individuals’ knowledge and awareness was the most common intervention [5].

I will not delve into individuals’ experiences with vaccines, beliefs about health, knowledge, trust in the health system, perceived risk/benefit, beliefs, or religion, among others. My goal is to bring to the discussion the vaccine patent system that has been mostly absent in the analysis of vaccine hesitancy causal factors.

In this 2023 Vaccines issue, other authors will cover individual determinants of the complex system of vaccine hesitancy and adherence. I have chosen to focus my commentary on the vaccine market. The scope of my commentary is the behavior of the industry, using the cases of HPV and COVID-19 vaccines as examples. My aim is to demonstrate that most of the research and development was supported by public funds in both cases, but, due to the patent system, they became a monopoly of the respective industries that patented them [6]. As a result, the new owners of the vaccine intellectual property determined the price and distribution of those vaccines. This resulted in an unfair distribution of the COVID-19 vaccines, leaving the poorer populations underserved during the worst months of the pandemic [7].

During COVID-19 pandemic, there were also examples of fairness and public health commitment. The Oxford AZ vaccine serves as just one example of how public investment resulted in a low-cost vaccine that benefited the public [8]. Additionally, the World Health Organization (WHO) played a key role in coordinating vaccine production and distribution with governments and international donors to reach the populations most in need. However, we must identify and correct the structural factors to not depend only on the goodwill of governments or private donors when a new pandemic arrives.

The vaccines presented here were developed based on sound science and technology, such virus-like particles (VLP) and mRNA platforms targeting Human Papilloma Virus (HPV) and SARS-CoV-2. Both vaccines were met with strong public hesitancy, which, in part, could be attributed to vaccine market opacity.

## 2. Vaccines Industry and Public Health

The technological revolution and the growth of the vaccine industry today traded on the stock market have profoundly modified the significance of vaccines for society. Pfizer’s stock was initially listed on the New York Stock Exchange in 1942 [9], whereas Merck’s stock joined in 1970. Thus, 80 years ago, vaccines initiated their transition from a public health good to a market good. The first vaccine producers in 1700 conducted basic research, produced the vaccines, and did the testing, sometimes first testing on themselves; they did not patent their products. Their primary objective was to halt the propagation of deadly epidemics such as smallpox [10]. Presently, vaccine producers are typically board members of large pharmaceutical companies, supported by a team of scientists who explore the most promising research findings from universities or small biotechnology firms to work from there. The pharmaceutical companies complete very efficiently the translation process from the laboratory to the population, being a key player in the vaccine production chain. One of the problems here is that the large social investment behind the vaccines does not enter into the equation when the companies calculate the prices they will charge to society. Thus, the price of the new vaccines is many times higher than the production costs, which the companies attribute to their investment in research and development, as we saw before this is not their main cost component. In addition to ignoring society’s ownership of most discoveries and innovations when deciding vaccine costs, these companies do not operate under free market competition [11], and then the price is not regulated by the market. In the pharmaceutical market, the competition is limited by potent Intellectual Property (IP) regulations signed by almost all developed and most low- and middle-developed countries [12]. Thus, these companies are one of the most profitable industries in the world [13]. They decide which vaccine to develop, its price, and its distribution. 

The AIDS epidemic created a dramatic situation when the pharmaceutical industry did not produce antiretroviral drugs affordable for African patients, but, based on their IP rights, they impeded the local production at one tenth of its price. This created a health gap between the quality of life and survival of AIDS patients and families in Africa versus those in the USA or Europe who were returning to their normal life thanks to the antiretroviral therapies [14]. This health inequality of the pharmaceutical market was repeated with the new Human Papilloma Virus (HPV) and again with the COVID-19 vaccines.

The Agreement on Trade-Related Aspects of Intellectual Property Rights (TRIPS) is an international legal agreement among all the member nations of the World Trade Organization. It establishes minimum standards for the regulation by national governments of different forms of intellectual property (IP). TRIPS was established in 1990 and is administered by the WTO [15]. It has some “flexibilities”, almost impossible to make operative in real life. The COVID-19 pandemic fueled critical evaluation, and many authors proposed a profound review of the Agreement [15,16,17,18].

Organizations such as the World Health Organization (WHO), COVID-19 Vaccine Global Access (COVAX), the Global Alliance for Vaccines and Immunization (GAVI), the Coalition for Epidemic Preparedness Innovations (CEPI), and non-profit entities like the Bill & Melinda Gates Foundation have endeavored to address these public health inequities. Unfortunately, these efforts found opposition among the industry and some governments where these industries are located [19]. Under significant public opinion pressure, one month before WHO declared the end of the COVID-19 pandemic, on 30 March 2023, MODERNA reported an agreement with the Republic of Kenya. With the auspices of the US government, Moderna will establish in Kenya an mRNA manufacturing facility that could produce 500 million doses of mRNA vaccine per year [20]. I hope the details of the agreement will secure that the benefits are primarily for the African population and are sustainable in time. This would increase the confidence of the public in vaccines and in pharmaceutical companies.

Unfortunately, at the peak of the COVID-19 pandemic, the strong opposition of the industry supported by the World Trade Organization (WTO) did not permit mRNA SARS-CoV-2 vaccines to be produced in India or South Africa when they were most needed [19,20,21]. It made very clear the opposite goals between the industry and the population’s health [7]. In addition to their lack of attention to people’s needs, the pharmaceutical companies have been found guilty many times of lobbying decision-makers to regulate for their benefit. I will present below the case of the HPV vaccine, highly covered by the press, where the industry through women’s organizations pushed a law for compulsory vaccination soon after the HPV vaccine approval by FDA.

The inequity public health crisis, brought by the IP during the AIDS epidemic and now in the COVID-19 Pandemic, demonstrated the failure of the companies in responding to public health global needs. The persistence of pockets of the virus, the emergence of virus variants, and the public’s growing vaccine hesitancy have not affected the earnings of the companies. They were the industry with the highest economic growth during the COVID-19 pandemic.

## 3. HPV Vaccine

### 3.1. Discovery of HPV Virus as Cause of Human Cancers

The Human Papillomavirus IARC’s Monograph of 2005 offers a summary of the historical studies that led to the discovery of HPV vaccines. The initial investigation of HPV was carried out in Brazil in 1920 in bovine warts, reminiscent of the origin of the first vaccine 225 years ago, the smallpox vaccine obtained from the cowpox virus. In 1963, it was reported that Bovine Papillomavirus BPV had the ability to transform cells in culture. Between 1970 and 1982, research efforts focused on identifying the segments of the BPV genome responsible for cell transformation; its complete sequence was reported in 1982. In parallel, since 1930, Peyton Rous studied the cotton rabbit papillomavirus at Rockefeller University, obtaining the 1961 Nobel Prize for demonstrating this papillomavirus-induced squamous-cell carcinoma in rabbits. In humans, Lewandowsky in 1922 described that epidermodysplasia verruciformis could progress to squamous-cell carcinomas, later associated with HPV 5. HPV 16 was cloned in 1983 and HPV 18 in 1984, and they were found to be responsible for 70% of cervical cancers. In 1985, the viral genes E6 and E7 were identified as key players in the process of carcinogenesis. In 2008, Harald zur Hausen won The Nobel Prize for his discovery of human papilloma viruses causing cervical cancer.

Large-scale epidemiological studies conducted in the early 1990s identified high-risk HPV types as the primary risk factors for cervical, anogenital, and oropharyngeal cancers. By 2004, the sequence of 100 HPV genotypes was completed (IARC 2005). These efforts represented more than a century of research conducted by hundreds of scientists in various universities, research centers, and international organizations worldwide, with funding predominantly provided by governments [22].

### 3.2. Discovery of Virus-like Particles (VLPs) as Prophylactic HPV Vaccines

In 2002, Harald zur Hausen published a concise review of papillomavirus history [23]. In 1991, at Queensland University of Technology in Australia, Zhou J. reported the spontaneous formation of Virus-Like Particles (VLPs) through the expression of L1, the viral capsid protein, and suggested their potential use as a vaccine [24]. In the same year, Meneguzzi at INSERM in France reported therapeutic results in rats with a vaccinia recombinant expressing E6 or E7 [25]. In 1995, there were papillomavirus vaccinations in dogs [26] and cottontail rabbits [27]. In 2001, was the first report of VLPs’ immunogenicity in humans [28].

In 2006, the first HPV vaccine (Gardasil TM) was authorized for use in the population [29]. Currently, 125 countries have introduced the HPV vaccine into their national immunization programs for girls, and 47 countries have also implemented it for boys. A recent comprehensive meta-analysis conducted in developed countries demonstrated the prolonged effectiveness of HPV vaccination in protecting against HPV infection, anogenital warts, and, most importantly, cervical neoplasia, which serves as a proxy for cervical cancer. The study also revealed a high level of herd immunity against HPV infection among men and older women, even in populations not included in the vaccination programs [30]. Furthermore, the analysis confirmed the absence of severe adverse events or adverse pregnancy outcomes in recipients of HPV vaccines [31].

In 2006, the National Institutes of Health (NIH) of USA declared that the nearly two decades of work of researchers at NIH’s National Cancer Institute (NCI) led to the development of the technology upon which the newly approved HPV vaccine (Gardasil TM) is based [29].

Unfortunately, the high prices set by the companies have resulted in a significant disparity in vaccination coverage and cervical cancer incidence and mortality between high- and low-income countries. In 2020, there were an estimated 604,127 cases of cervical cancer and 341,831 related deaths worldwide. Mortality rates ranged from 1.0 (0.8–1.2) in Switzerland to 55.7 (47.7–63.7) in Eswatini (formerly Swaziland) in Africa. That year, the WHO launched the global Cervical Cancer Elimination Initiative that requires 90% of girls to be vaccinated by the age of 15 years by 2030 [32].

## 4. HPV Vaccine, the State, the Industry, and Public Hesitancy

The HPV vaccine faced concerns even before its approval by the FDA. As Colgrove described in 2006, there were philosophical, political, scientific, and ideological reasons for parents to oppose mandatory vaccination. People´s reasons for rejecting vaccines were: “preferred alternative or natural healing, libertarian opposition to state power, mistrust of pharmaceutical companies, and belief that children receive more shots than are good for them” [26]. Religious groups were concerned of a vaccine against sexually transmitted diseases will oppose their teachings of abstinence. Bioethicists argue that they are skeptical about a compulsory vaccination law since HPV is not haphazardly transmitted, an infected person will not casually infect another [33].

In this context of worldwide increasing hesitancy against the compulsory HPV vaccination, a Texas governor made an executive order to make Gardasil mandatory in his State. Soon after, the press reported Merck´s Advocacy for the Gardasil vaccine, affecting the governor´s chief of staff. The Governor had to rescind his order, and Merck declared they stopped lobbying for Gardasil [34,35]. A 2007 health poll by the Kaiser Family Foundation found that fewer than 18 percent of Americans “say they can trust what pharmaceutical companies say in their ads most of the time”. Some 70% “agree that drug companies put profits ahead of people” [34,35]. This widely publicized case exemplifies how this lobbying activity erodes public trust in both the companies themselves and politicians [36]

## 5. COVID-19 Vaccines

### Discovery of the mRNA Vaccines

By February 2021, while Israel had more than 78 cumulative COVID-19 vaccination doses administered per 100 people, Cambodia, Pakistan, Mauritius, Albania, Ecuador, Guyana, and Bolivia had less than 0.1 doses administered [21]. The mRNA vaccines carry the hidden costs of sixty-two years of publicly funded research, as aptly summarized by S. Pascolo [37]. It began with the discovery of mRNA in 1961 by F. Jacob and J. Monod at the Pasteur Institute, Paris. This was followed by mechanistic and animal studies at the Universities of Philadelphia, Harvard, and Illinois, which showed that these short RNAs can induce proteins, to then find that human cells infected with mRNA can express rabbit proteins, and the production in 1982, synthetic mRNA un vitro, at Harvard University, which opened the possibility of designing mRNA for human use [37]. In 1983 studies at Ghent University in Belgium, in 2002 at University of Warsaw, Poland, and, in 2017, the biotechnological company Trilink in California secured in vitro mRNA functionality and mRNA production. Scientists from the Institut National de la Santé in France in 1993 produced the first mRNA vaccines. In 2000, scientists in Germany founded the company CureVac to develop cancer therapies based on mRNA. They soon initiated the use of in vitro mRNA vaccines through direct injection, with Pascolo himself being the first human to receive an mRNA vaccine. Subsequently, melanoma patients were vaccinated, paving the way for mRNA vaccine therapies [37]. In Parallel, scientists at Tubingen University working on mRNA vaccines for cancer funded in 2008 the company BioNTech, becoming the only two companies in the field worldwide [37]. In 2020, BioNTech tested the COVID-19 vaccine for the first time in humans, comparing a vaccine that encoded the receptor binding domain of the spike, with another that encoded for the whole spike protein; it was a multicenter study in Germany, later tested in the USA by BioNTech and Pfizer together and commercialized as Comirnaty. This was followed by Moderna and SpikeVax. These successes were possible due to the 61 years of hidden public investment [37] but also by the large public funds allocated the to the mRNA vaccines’ development.

## 6. Discussion

There is a broad range of determinants of vaccine hesitancy from the vaccine or vaccination itself (cost, mode of administration, schedule, and reliability in the supplier, among others), from the individual and the social group (beliefs, attitude, knowledge, and trust in the health personnel), and from the political and economic context including the perception of the pharmaceutical industry [4]. I focused on the latter, which is the least studied and addressed [5].

I presented two different vaccines, one to prevent the infection with an oncogenic virus to be applied in children, the other against a respiratory virus with high pandemic potential to be used by the whole population. Both are excellent, safe, and based on sound science, yet both have encountered strong public hesitancy.

I demonstrated that the research, discovery, and innovation were performed in large part with public funds in universities and research centers in both cases. In both, the industry entered the final stage to produce, test, and distribute large quantities of vaccines. In both cases, the costs of the current vaccines are hidden, but scientists estimate they are many times lower than their market price [11].

The lack of transparency in this market and the cases of payments to politicians or decision-makers are behind the anti-vaccine movements that are strongly supported by populist leaders worldwide [34,35,38,39]. As Gugushvili A stated, “It is ‘the people’ versus the political and financial elites, with medical and scientific experts seen as among those who are deemed elitist, speaking a different, inaccessible language and entwined with big business and pharma as well as politics” [40].

In the case of the COVID-19 pandemic, the industry did not have the capacity to produce the vaccine for the whole world. Yet they did not allow the COVID-19 vaccine to be produced in Brazil, India, Africa, or Canada, laboratories that had the conditions and were ready to produce COVID-19 vaccines [15,41].

The WHO tried to solve the emergency by creating the COVAX, but failed. As K. Storeng put it in 2021, “The COVID-19 Vaccines Global Access Facility (COVAX) a public–private partnership … embraces the current intellectual property regime as a necessary driver of innovation. Thus, it has allowed pharmaceutical companies to keep vaccine contracts and prices secret, and it has defended its resistance to sharing vaccine technology, despite globally limited vaccine supply. …Although the WHO has been an outspoken supporter of the waiver, the UK, Norway, Switzerland and the European Union (EU) have blocked the effort for over a year. As the EU Ambassador to the African Union put it, “Europe Supports COVAX, not the TRIPS waiver”” [42].

To solve the significant gap between COVID-19 vaccine coverage in Africa and the rest of the world, numerous initiatives of international agencies, vaccine producers, and local governments collaborated to increase COVID-19 vaccine availability in Africa [43,44]. This crisis has brought attention to the need for a vaccine production infrastructure accessible to LMICs (www.devex.com). There have been many initiatives to increase research and development in the pharmaceutical industry for other urgent public health needs. Some authors have proposed government economic incentives to advance research and development into antibiotics for multi-resistant bacteria [45,46], cancer drugs [47,48], medicines [49], and malaria vaccines [50]. These incentives may include research grants, rewards for research advancements, and facilitating access to knowledge through open-source platforms [49]. However, a critical challenge, as highlighted by Fatima Suleman, is that these incentives should include the requirement for fair pricing rather than solely focusing on profit margins [49].

The vaccine crisis created by the Intellectual Property regulation maintains, until today, the important numbers of people without the COVID-19 vaccine.

In 2019, the WHO passed a resolution for a more transparent market for vaccines and other products; to date, it has not been implemented. For COVID-19 vaccines, governments provided public funding for research and development, lowering most of the risks for the industries. Unfortunately, governments did not request the companies’ transparency of their costs to translate them into an affordable price of the vaccine [51].

During a public health emergency, when all society moves their usual restrictions, pharma should have also reduced their expected earnings and facilitated their know-how to low-income countries where they could not buy their vaccines.

This unfair and opaque market is a contributing factor to public vaccine hesitancy. In order to restore confidence in vaccines and better prepare for future pandemics, it is imperative to initiate a change in the patent system governing these essential public health goods. This change should prioritize transparency, independence, and fairness.

## 7. Conclusions

I propose that the road to the recovery of public trust requires:Enhancing transparency within the vaccine market. Estimating and discounting the sunk costs of training and research behind each vaccine, which are paid for by the public, from those costs paid by the industry.Evaluating the outcomes of the patent system created to secure innovation, and critically assessing where discoveries and innovations are taking place.Reviewing the patenting processes, ensuring the origin of the innovations being patented and guaranteeing that the flexibility process will work in the next pandemic.Engaging health professionals and the scientific community in supporting the vaccine from the beginning of the scientific search, discovery, testing, to its transfer to the industry.Warranting that the scientists who present the data supporting the vaccines and those who promote vaccinations are absolutely independent from the vaccine companies.Avoiding direct or indirect payments (education and conferences) from the vaccine companies to doctors, politicians, regulators, or public health administrators.Establishing a vaccine observatory within the WHO with access to information on production costs, the status of vaccine development, and vaccine trials to advise the countries and inform the public.

## Data Availability

Not applicable.

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
