# Peer review of "How Did We Get Here: The Best Vaccines Ever Facing the Highest Public Hesitancy?"

_vaccines, 2023, doi:10.3390/vaccines11081323_

Round 1

Reviewer 1 Report

In this manuscript, the author gives a review of vaccines to the public health basing on the case of HPV and COVID-19 vaccines. However, critical questions and reflections on vaccine hesitancy in the public are not sufficiently addressed, and are not discussed with the actual application of HPV and COVID vaccines. And the descriptions of HPV and COVID vaccines are not comprehensive enough.

Author Response

Regarding the critique,

I created a new paragraph, introduction, explaining the scope of my review.

Please see below.

Most reports related to vaccine hesitancy have approached it from individual subjective factors, consequently recommending individual-based interventions to increase vaccine acceptance, such as,  the use of commercial and social marketing principles [18], identifying vaccine hesitancy population subgroups [19], or improving health communication using novel tools and strategies [20]. Only one study included in its Hesitancy Matrix “perception of the pharmaceutical industry” as one of the contextual influences in vaccine hesitancy [21]. A review of vaccine hesitancy interventions from 1996 to 2013 found that targeting individuals' knowledge and awareness was the most common intervention [1].

I will not delve into individuals' experiences with vaccines, beliefs about health, knowledge, trust in the health system, perceived risk/benefit, beliefs, or religion, among others. My goal is to bring to the discussion the vaccine patent system which has been mostly absent in the analysis of vaccine hesitancy causal factors.

In this 2023 Vaccines issue, other authors will cover individual determinants of the complex system of vaccine hesitancy and adherence. I have chosen to focus my commentary on the vaccine market. The scope of my commentary is the behavior of the industry, using the cases of HPV and COVID-19 vaccines as examples. My aim is to demonstrate that in both cases, most of the research and development was supported by public funds, but due to the patent system, they became a monopoly of the respective industries that patented them [15]. As a result, the new owners of the vaccine intellectual property determined the price and distribution of those vaccines. This resulted in an unfair distribution of the COVID-19 vaccines, leaving the poorer populations underserved during the worst months of the pandemic.

During COVID-19 pandemic, there were also examples of fairness and public health commitment. The Oxford AZ vaccine serves as just one example of how public investment resulted in a low-cost vaccine that benefited the public [22]. Additionally, the World Health Organization (WHO) played a key role in coordinating vaccine production and distribution with governments and international donors to reach the populations most in need. However, we must identify and correct the structural factors to not depend only in the goodwill of governments or private donors when a new pandemic arrives.

 The vaccines presented here, were developed based on sound science and technology, virus-like particles (VLP), and mRNA platforms, targeting Human Papilloma Virus (HPV) and SARS-CoV-2. Both vaccines were met with strong public hesitancy, which in part could be attributed to vaccine market opacity.

  1. Regarding the recommendation to improve the results,

I am not reporting a particular study but presenting an opinion partly based on literature (both grey and peer-reviewed) and my own experience with both vaccines. Regarding the HPV vaccine, in 2014, as a member of the vaccines advisory committee of the Chilean Ministry of Health (MoH), I was responsible for designing the HPV first vaccine Program. As for COVID-19 vaccines, from 2020 to date, as a member of the Response to the COVID-19 Pandemic advisory board at the MoH, I have been involved in advising on the COVID-19 vaccine intervention. I present historical data about vaccine research, development, and marketing, which I propose affect people's confidence, thus creating vaccine hesitancy. Therefore, my article does not have a results section. I have added a clarification of what to expect in the new paragraph called "Introduction."

Reviewer 2 Report

The commentary is very interesting ang well documented. 

I have no particular comment except to make sure to standardize the font. 

Author Response

Response to reviewer 2:

Thank you very much for your encouraging evaluation.  We will check the fonts of the final proof of the article.

Reviewer 3 Report

This is a valuable contribution to the ongoing hesitancy of the public to vaccinate at least partially based on the assumption that industry sells the products at unfair high prices and does not allow production at low cost by others.

The problem so far is that critical statements without justification are made such as:

·      Payments to politicians or decision-makers, but no evidence or details are provided.

·      The high “unfair” market price is repeatedly addressed, but not exemplified. Basic research has been supported over years by public funding, but no information on efforts of industry to produce the vaccines, organize safety assessment and final approvement by the competent agencies is provided. 

 Although one may agree or disagree with the Conclusions the arguments are a basis for further discussions.

Author Response

Regarding the background, methods, and results, as I explained to reviewer 1, please see the new introduction.

For the case of payments to politicians, there were four references listed in my reference list that unfortunately were not cited in the text. Now, this has been corrected. Here are the original references:

  • Angell, M., The truth about the drug companies: how they deceive us and what to do about it. 2004: First edition. New York: Random House, [2004] ©2004.
  • Wouters, O.J., Lobbying Expenditures and Campaign Contributions by the Pharmaceutical and Health Product Industry in the United States, 1999-2018. JAMA Intern Med, 2020. 180(5): p. 688-697.
  • Merck's Murky Dealings: HPV Vaccine Lobby Backfires. 2007. May 18, 2023. Available from: https://www.corpwatch.org/article/mercks-murky-dealings-hpv-vaccine-lobby-backfires.
  • Tanne, J.H., Texas governor is criticized for the decision to vaccinate all girls against HPV. BMJ, 2007. 334(7589): p. 332-3.

In fact, I made a strong statement based on the Canada-Bolivia case reference reported by Crombie, which unfortunately was not included in my reference list. Now, I have added it:

Crombie J. Intellectual property rights trump the right to health: Canada’s Access to Medicines Regime and TRIPs flexibilities in the context of Bolivia’s quest for vaccines. Journal of Global Ethics, 2021, Vol. 17, No. 3, 353–366. https://doi.org/10.1080/17449626.2021.1993452

Additionally, there was an omission of citing Kohler J in the text, although it was included in my reference list:

 Kohler J, Wong A, Tailor L. Improving Access to COVID-19 Vaccines: An Analysis of TRIPS Waiver Discourse among WTO Members, Civil Society Organizations, and Pharmaceutical Industry Stakeholders. Health Hum Rights. 2022 Dec;24(2):159-175. PMID: 36579316; PMCID: PMC9790937.

Furthermore, the statement about the "unfair market price" is based on the analysis of the two vaccines presented in this commentary. Both vaccines received substantial public investment in their research and development, which did not result in a return to the public. Once the companies obtain intellectual property rights, they have the power to determine the price in a monopolistic market.

The cost and pricing of COVID-19 vaccines were described by Light DW 2021, which is included in my reference list and now cited in the text:

  • Light, D.W. and J. Lexchin, The costs of coronavirus vaccines and their pricing. Journal of the Royal Society of Medicine, 2021. 114(11): p. 502-504.

Many authors in my reference list describe this unfairness (Altindis 2022, Angell M 2004, Arguedas-Ramirez 2022, Chattu VK 2021 b, Ledford 2021). Crombie 2021 specifically describes how the companies impeded the use of patent waivers to benefit Bolivia (Crombie 2021).

Reviewer 4 Report

Thank you for this paper to stimulate debate in this important area. However - the paper contains a number of inaccuracies/ left out information to make a more balanced paper. 

These include the following:

A) COVID-19 

a) There is no mention of the Oxford AZ vaccine. The research was predominantly funded by the British Government, with AZ principally making the vaccine available at cost - hence appreciably lower price that e.g. the Pfizer vaccine. This is called the 'decoupling model' which has been suggested for e.g. new antibiotics since these are likely to be restricted in their use - hence limited commercial interest (e.g. Anderson M, Mossialos E. Incentivising antibiotic research and development: is the UK's subscription payment model part of the solution? Lancet Infect Dis. 2020;20(2):162-3 and Barlow E et al. Optimal subscription models to pay for antibiotics. Soc Sci Med. 2022;298:114818 - among others) - In addition - suggested for new cancer treatments as typically R & D funded by governments/ charities before the Industry become involved leading to some very high/ unjustified prices - due to the emotive nature of the disease area - leading to calls for fairer prices (discussed in e.g. Workman P et al. How Much Longer Will We Put Up With $100,000 Cancer Drugs? Cell. 2017;168:579-83; Suleman F et al. New business models for research and development with affordability requirements are needed to achieve fair pricing of medicines. Bmj. 2020;368:l4408 and Godman B et al. Potential approaches for the pricing of cancer medicines across Europe to enhance the sustainability of healthcare systems and the implications. Expert Rev Pharmacoecon Outcomes Res. 2021;21:527-40).  Interestingly, the Oxford/ AZ vaccine received a great deal of unjustified criticism initially including lack of effectiveness in the elderly and possible clots, etc. However - no differences subsequently seen vs. other vaccines as more data became available - one wonders if other companies were casting false claims over the Oxford/ AZ vaccine as considerable cheaper than e.g. Pfizer vaccine without the storage/ transport difficulties - Hippisley-Cox et al. Risk of thrombocytopenia and thromboembolism after covid-19 vaccination and SARS-CoV-2 positive testing: self-controlled case series study. Bmj. 2021;374:n1931. Possible refs include - https://www.gov.uk/government/news/one-year-anniversary-of-uk-deploying-oxford-astrazeneca-vaccine; https://www.gov.uk/government/news/funding-and-manufacturing-boost-for-uk-vaccine-programme; https://www.bbc.co.uk/news/health-55041371 

b) This is not the first time when high priority R & D has been sponsored by the British Government. We saw this with the Recovery trial for re-purposed medicines given issues regarding HCQ - which despite the initial hype not effective - Horby P et al. Effect of Hydroxychloroquine in Hospitalized Patients with Covid-19. N Engl J Med. 2020;383:2030-40. Similar for lopinavir - Lopinavir-ritonavir in patients admitted to hospital with COVID-19 (RECOVERY): a randomised, controlled, open-label, platform trial. Lancet. 2020;396:1345-52. Only dexamethasone shown to have some effect - Horby P et al. Dexamethasone in Hospitalized Patients with Covid-19. N Engl J Med. 2021;384(8):693-704.

c) There is no mention of e.g. the Chinese COVID-19 vaccines - which have been made available in e.g. Africa - discussed in e.g. Ogunleye OO et al. Coronavirus Disease 2019 (COVID-19) Pandemic across Africa: Current Status of Vaccinations and Implications for the Future. Vaccines. 2022;10:1553 

d) There was also a tie up between J & J and Aspen in South Africa to help move production of the Vaccines to Africa. However - not taken up leading to potential closure of the plant - https://www.reuters.com/business/healthcare-pharmaceuticals/safricas-aspen-covid-19-vaccine-plant-risks-closure-after-no-orders-executive-2022-05-01/; https://www.jnj.com/johnson-johnson-announces-landmark-agreement-to-enable-its-covid-19-vaccine-to-be-manufactured-and-made-available-by-an-african-company-for-people-living-in-africa; https://www.devex.com/news/prospects-for-local-manufacturing-of-covid-19-vaccines-in-africa-102300

B) HPV vaccine 

a) There have been concerns with the vaccine - nothing to do with Pharma, etc. e.g. Smith LM et al Effect of human papillomavirus (HPV) vaccination on clinical indicators of sexual behaviour among adolescent girls: the Ontario Grade 8 HPV Vaccine Cohort Study. Cmaj. 2015;187:E74-e81 and Ambali RT et al. Indepth Interviews' on Acceptability and Concerns for Human Papilloma Virus Vaccine Uptake among Mothers of Adolescent Girls in Community Settings in Ibadan, Nigeria. J Cancer Educ. 2022;37:748-54 (although costs come in here). Fathers are also a key player as well influencing uptake (not discussed) - Kolek CO et al. Impact of Parental Knowledge and Beliefs on HPV Vaccine Hesitancy in Kenya;Findings and Implications. Vaccines. 2022;10:1185 

C) Brazil - FioCruz

a) We see the Brazilian Government along with Bill and Melinda Gates funding research into a malaria vaccine (discussed in e.g. Labis da Costa MJ et al. Willingness to pay for a hypothetical malaria vaccine in Brazil: a cross-sectional study and the implications. J Comp Eff Res. 2022;11:263-74

b) Fiocruz in Brazil is also involved in producing many biological medicines, etc., for the public health system in Brazil - not mentioned 

D) Next steps - 2 distinct events - one about obtaining realistic/ fair prices especially if the R & D has been sponsored by governments/ charities following the Oxford/ AZ example for COVID-19 and decoupling suggestions for new antibiotics. Secondly - addressing vaccine hesitancy - because this contagious across vaccines especially with the rise in social media activities/ misinformation and its impact    

Worth just checking this again

Author Response

Thank you very much for your thoughtful review and the recommended readings. I agree with all your comments and have made the necessary revisions to address them. I understand that certain topics cannot be further developed in this commentary due to space limitations.

A)

  1. a) I agree with the reviewer that the Oxford AZ vaccine was a successful model of how public investment returned to the public and the world. I have added this information to my manuscript. Describing the teachings of the AZ vaccine would require a full article, but I mention it in the Introduction, including some of the references suggested by the reviewer.
  2. b) As mentioned above, I chose to focus my commentary on the market-patent aspect of the problem.
  3. b) Thank you. Since this is a general commentary aimed at raising awareness of the problem rather than proposing economic solutions, I have included a paragraph in the discussion to mention the incentives proposed by Fatima Suleman, 2020.
  4. c) Thank you for the reference. It is a very interesting report with excellent recommendations applicable to Africa and many other low- and middle-income countries (LMICs). I have included it in the introduction.
  5. d) Thank you for these suggestions. I have added them to the discussion:

Discussion:

 To solve the significant gap between COVID-19 vaccine coverage in Africa and the rest of the world, numerous initiatives of international agencies, vaccine producers, and local governments collaborated to increase COVID-19 vaccine availability in Africa [49]. This crisis has brought attention to the need for a vaccine production infrastructure accessible to LMICs [www.devex.com]. There have been many initiatives to increase research and development in the pharmaceutical industry for other urgent public health needs. Some authors have proposed government economic incentives to advance research and development in antibiotics for multi-resistant bacteria [50, 51], cancer drugs [52, 53], medicines [54], and malaria vaccines [55]. These incentives may include research grants, rewards for research advancements, and facilitating access to knowledge through open source platforms [54]. However, a critical challenge, as highlighted by Fatima Suleman, is that these incentives should include the requirement for fair pricing rather than solely focusing on profit margins [54].

  1. B) HPV Vaccine:

In the Introduction, I have now made explicit the boundaries of my commentary. Thank you for the suggestion.

  1. C) Brazil:

I have included examples of efforts to overcome limitations in the current pharmaceutical market, citing some of the suggested references. Thank you. Please see the new paragraph in the discussion.

D)

I have addressed this point in a new paragraph in the discussion, citing some of the references you kindly suggested. Thank you.

Round 2

Reviewer 1 Report

The authors analyzed the phenomenon of public vaccine hesitancy through the COVID-19 vaccine and HPV vaccine. Although mRNA and VLP vaccines play a very good role, some people are still wary of vaccines. However, I still have some major concerns and suggestions that need to be addressed.

1. There is a need to discuss whether the core issue of public concern about vaccines is the safety of vaccines or the protective effect of vaccines and social communication.

2. There are still other kinds of vaccine strategies against SARS-CoV-2 and HPV, and their impact on reducing public hesitancy needs to be discussed.

3. The authors do not provide data on the proportion of people with vaccine hesitancy or the impact on viral prevalence.

4. Relevant data and analysis should be provided on the beneficial effects of vaccines to the public hesitancy.

Author Response

The authors analyzed the phenomenon of public vaccine hesitancy through the COVID-19 vaccine and HPV vaccine. Although mRNA and VLP vaccines play a very good role, some people are still wary of vaccines. However, I still have some major concerns and suggestions that need to be addressed.

Re: Thank you for your comments bellow, I answer each of them: 

  1. There is a need to discuss whether the core issue of public concern about vaccines is the safety of vaccines or the protective effect of vaccines and social communication.

Re: As I explained in the introduction, I opted to present an opinion about the role of the vaccine market in vaccine hesitancy, that the full journal issue will be devoted to this topic, thus I add a piece to a much larger puzzle. I did not aim to explore safety, efficacy, or social communication.

  1. There are still other kinds of vaccine strategies against SARS-CoV-2 and HPV, and their impact on reducing public hesitancy needs to be discussed.

Re: Of course, there are other public health measures like isolation, face masks, quarantines for COVID19 and postponement of sexual initiation, for HPV. But I did not aim at discussing the role of these vaccines in controlling the infections, but to describe the source of the funding for the science behind these to vaccines and the insufficient recognition of this fact by the current patent system.

  1. The authors do not provide data on the proportion of people with vaccine hesitancy or the impact on viral prevalence.

Re: As I explained in my answer to comment #1: these questions are out of the scope of my commentary.

  1. Relevant data and analysis should be provided on the beneficial effects of vaccines to the public hesitancy.

Re: I do not quite get the question; I assume you mean how an effective vaccine will help to decrease public hesitancy. Sure, good results will help very much in decreasing public hesitancy, as we saw with the Chinese vaccines acceptability during the pandemic. But that will also divert the focus of my commentary.

Reviewer 3 Report

The author has considered my comments and added the necessary information. I have no further comments.

Author Response

I thank very much to this reviewer whose suggestions help me to improve the manuscript.

Reviewer 4 Report

Thank you - no further comments to make.

Author Response

I thank very much to this reviewer whose commentaries and suggestions help so much to improve my manuscript.